# Polarized Baryon Production in Heavy Ion Collisions: An Analytic Hydrodynamical Study

**Bálint Boldizsár, Márton I. Nagy and Máté Csanád *** 

Department of Atomic Physics, Eötvös Loránd University, Pázmány P. s. 1/A, H-1117 Budapest, Hungary;
boldizsar.balint@hotmail.com (B.B.); nmarci@elte.hu (M.I.N.)

* Correspondence: csanad@elte.hu

**Abstract:** In this paper, we utilize known exact analytic solutions of perfect fluid hydrodynamics to analytically calculate the polarization of baryons produced in heavy-ion collisions. Assuming local thermodynamical equilibrium also for spin degrees of freedom, baryons get a net polarization at their formation (freeze-out). This polarization depends on the time evolution of the Quark-Gluon Plasma (QGP), which can be described as an almost perfect fluid. By using exact analytic solutions, we can thus analyze the necessity of rotation (and vorticity) for non-zero net polarization. In this paper, we give the first analytical calculations for the polarization four-vector. We use two hydrodynamical solutions; one is the spherically symmetric Hubble flow (a somewhat oversimplified model, to demonstrate the methodology); and the other solution is a somewhat more involved one that corresponds to a rotating and accelerating expansion, and is thus well-suited for the investigation of some of the main features of the time evolution of the QGP created in peripheral heavy-ion collisions (although there are still numerous features of real collision geometry that are beyond the scope of this simple model). Finally, we illustrate and discuss our results on the polarization.

**Keywords:** hydrodynamics; heavy ion collisions; polarization

---

## 1. Introduction

Our aim is to give analytical results for the polarization four-vector of massive spin 1/2 particles produced in heavy-ion collisions from hydrodynamical models. The motivation for this work was the recently observed non-vanishing polarization of $\Lambda$ baryons at the STAR (Solenoidal Tracker at the Relativistic Heavy Ion Collider) experiment [1,2] that hints at local thermal equilibrium also for spin degrees of freedom in the Quark Gluon Plasma (QGP) produced in heavy-ion collisions. The assumption of thermal equilibration for spin is at the core of the current understanding of polarization of particles produced from a thermal ensemble (such as the QGP), and almost all studies aimed at describing it in terms of collective models utilize the formula derived from this assumption by Becattini et al. [3].

Although many numerical hydrodynamical models do indeed predict non-zero polarization of produced spin 1/2 particles [4–7], a clear connection between the initial state, the final state, and the observable polarization is to be expected from analytical studies, on which topic we do the first calculations here (to our best knowledge).

The observable quantities at the final state of the hydrodynamical evolution can be described by utilizing the kinetic theory. At local thermodynamical equilibrium, for spin 1/2 particles, such a description can be based on the the Fermi–Dirac distribution:

$$f(x,p) \propto \frac{1}{\exp\left(\dfrac{p_\mu u^\mu(x)}{T(x)} - \dfrac{\mu(x)}{T(x)}\right) + 1},\tag{1}$$

where $p_\mu$ is the four-momentum of the produced particle, and $u^\mu(x)$, $\mu(x)$, and $T(x)$ are the four-velocity, the chemical potential, and the temperature field of the fluid, respectively.

Assuming local thermal equilibrium for the spin degrees of freedom, for the space-time- and momentum-dependent polarization four-vector $\langle S(x,p)\rangle^\mu$ of the produced particles, the following formula is given in Ref. [3]:

$$\langle S(x,p)\rangle^\mu = \frac{1}{8m}\left(1 - f(x,p)\right)\varepsilon^{\mu\nu\rho\sigma} p_\sigma \partial_\nu \beta_\rho,\tag{2}$$

where $m$ is the mass of the investigated particle, and the inverse temperature field $\beta^\mu = u^\mu/T(x)$ is introduced. Here, $\varepsilon^{\mu\nu\rho\sigma}$ is the totally antisymmetric Levi-Civita-symbol, where the $\varepsilon^{0123} = 1$ convention is used. In this paper, we use this formula to calculate the polarization four-vector at the freeze-out from analytical, relativistic, hydrodynamical solutions.

The general consensus is that the appearance of polarization strongly depends on the rotation of the expanding QGP fireball. However, the Equation of State (EoS) of the QGP influences the rotation; thus, by measuring the polarization, we can get information about the EoS of the QGP. Analytic hydrodynamic calculations may provide special insight by yielding analytic formulas for the connections of the aforementioned physical quantities.

We investigate two hydrodynamical solutions: the spherically symmetric Hubble flow [8,9] and a rotating and accelerating solution (first reported in Ref. [10], then in a different context in [11]). We expect to obtain zero polarization in the case of the spherical symmetric Hubble-flow, as it has no rotation, so the study of this solution can be regarded as a simple cross-check of our methodology. The second one, however, being a rotating and expanding solution, could be a well-usable model of peripheral heavy-ion collisions, and it is expected that one gets non-zero polarization out of it. Thus, this rotating expanding solution constitutes the core point of the reported work.

## 2. Basic Equations and Assumptions

We use the $c = 1$ notation. Let us denote the space–time coordinate by $x^\mu \equiv (t, \mathbf{r})$, and the Minkowskian metric tensor by $g^{\mu\nu} = \mathrm{diag}(1, -1, -1, -1)$. The convention for the Levi-Civita symbol is $\varepsilon^{0123} = 1$. Greek letters denote Lorentz indices, and Latin letters denote three-vector indices. For repeated Greek indices, we use the Einstein summation convention. We denote the space dimension by $d$; this implies $g^\mu{}_\mu = d+1$. In reality, $d = 3$, but it is useful to retain the $d$ notation wherever possible, in order to see whether the reason for a specific numeric constant in the formulas is the dimensionality of space. The four-velocity of the fluid is $u^\mu = \gamma(1, \mathbf{v})$, where $\gamma = \sqrt{1 - v^2}$ is the Lorentz factor. The velocity three-vector is then $\mathbf{v} = u^k/u^0$. With $p^\mu$, we denote the four-momentum of a produced particle; we also use the three-momentum $\mathbf{p}$, whose magnitude we simply denote by $p$ (whenever there is no risk of confusion). The energy of the particle is denoted by $E$; the mass shell condition then reads as $E = \sqrt{p^2 + m^2}$, with $m$ being the particle mass.

The usability of hydrodynamics in heavy ion physics phenomenology relies on the assumption of local thermodynamical equilibrium of the matter. For describing particles with spin 1/2, we use the source function as written up in Equation (1). Hadronic final state observables can then be calculated

by integrating over the freeze-out hypersurface; for example, in the case of the invariant momentum distribution, the driving formula is

$$E\frac{\mathrm{d}N}{\mathrm{d}^3\mathbf{p}} = \int \mathrm{d}^3\Sigma_\mu(x)p^\mu f(x,p). \tag{3}$$

Here, $\mathrm{d}^3\Sigma_\nu$ is the three-dimensional vectorial integration measure of the freeze-out hypersurface, the appearance of which is the so-called Cooper-Frye prescription [12] for calculating the invariant momentum distribution. Of the two solutions (mentioned above) which we investigate in this work, in the case of the rotating and expanding accelerating solution, we also calculate the invariant momentum distribution, as this has not been done before.

The formula given in Ref. [3] for the polarization of spin $1/2$ particles, as written up in Equation (2), may be utilized for any given $\beta^\mu = u^\mu/T$ field that one gets from a given solution of the hydrodynamical equations. We are interested in calculating the polarization at the final state of the hydrodynamical evolution, so we must integrate the $\langle S(x,p)\rangle^\mu$ field over the freeze-out hypersurface. The formula to be analyzed further, that is, that for the observed polarization $\langle S(p)\rangle^\mu$ of particles with momentum $p$, thus becomes

$$\langle S(p)\rangle^\mu = \frac{\int \mathrm{d}^3\Sigma_\nu p^\nu f(x,p)\langle S(x,p)\rangle^\mu}{\int \mathrm{d}^3\Sigma_\nu p^\nu f(x,p)}, \tag{4}$$

as written up in, for example, [7]. For being able to perform analytical calculations, we had to make some assumptions. We used saddle-point integration, in which one assumes that the integrand is of the form $f(\mathbf{r})g(\mathbf{r})$, where $f(\mathbf{r})$ is a slowly changing function, while $g(\mathbf{r})$ has a unique and sharp maximum; then the integral can be calculated with a Gaussian approximation, as

$$\int \mathrm{d}^d\mathbf{r}\, f(\mathbf{r})g(\mathbf{r}) \approx f(\mathbf{R}_0)g(\mathbf{R}_0)\sqrt{\frac{(2\pi)^d}{\det \mathbf{M}}}, \quad \begin{array}{ll} \text{where} & \mathbf{M}_{ij} = \partial_i\partial_j g(\mathbf{r})\big|_{\mathbf{r}=\mathbf{R}_0}, \\ \text{and} & \partial_k g(\mathbf{R}_0) = 0; \end{array} \tag{5}$$

that is, $\mathbf{R}_0$ is the location of the unique maximum of $g(\mathbf{r})$, and $\mathbf{M}$ is the second derivative matrix.

Another assumption concerns the expression of $\langle S(x,p)\rangle^\mu$, Equation (2): if the exponent in the Fermi–Dirac distribution is large (i.e., phase space occupancy is small), we can use the Maxwell–Boltzmann distribution instead:

$$f(x,p) \ll 1 \quad \Rightarrow \quad f(x^\mu, p^\mu) = \frac{g}{(2\pi\hbar)^d}\exp\left(\frac{\mu(x)}{T(x)} - \frac{p_\mu u^\mu}{T(x)}\right). \tag{6}$$

Here, $g$ is the spin-degeneracy factor; for spin $1/2$ baryons, $g = 2$.

In high-energy heavy ion phenomenology (when the collision energy is high enough, say for collisions at the Relativistic Heavy Ion Collider or the Large Hadron Collider), the $\mu/T$ factor can be (and usually is) neglected; we use this approximation here[1]. With this, we have

$$f(x^\mu, p^\mu) = C_0 \exp\left(-p_\mu \beta^\mu(x)\right), \quad \text{where} \quad \beta^\mu(x) = \frac{u^\mu(x)}{T(x)}, \quad \text{and} \quad C_0 = \frac{g}{(2\pi\hbar)^d}. \tag{7}$$

---

[1]　The vanishing of $\mu$ can also be interpreted as an absence of a conserved particle number density $n$. All our conclusions would change only by a proportionality factor if we said $\mu/T = $ const instead of $\mu/T = 0$; if $\mu \neq 0$, we would have had to introduce $n$. Depending on the EoS (Equation of State) of the matter (one that also contains the conserved particle density $n$), one could write the $f(x,p)$ function in another form, where the normalization $\int \mathrm{d}p\, f(x,p) = n(x)$ is evident. For example, if one chooses an ultra-relativistic ideal gas, with $p = nT$, $\varepsilon = \kappa p$, with $\kappa = d$ as EoS, one has $\frac{g}{(2\pi\hbar)^d}e^{\mu/T} = \frac{n}{4\pi T^3}$. Indeed, in the solutions discussed below, $\mu/T = $const is satisfied, which means $n \propto T^d$, which is the well-known condition for an adiabatic expansion.

If the Maxwell-Boltzmann approximation is justified, it means that $f(x, p) \ll 1$ indeed, and Equations (2) and (4) then also become simpler:

$$\langle S(x, p) \rangle^{\mu} = \frac{1}{8m} \varepsilon^{\mu\nu\rho\sigma} p_{\sigma} \partial_{\nu} \beta_{\rho}, \tag{8}$$

and in the saddle-point approximation, the polarization of particles with momentum $p$ simply becomes

$$\langle S(p) \rangle^{\mu} \approx \frac{1}{8m} \varepsilon^{\mu\nu\rho\sigma} p_{\sigma} \partial_{\nu} \beta_{\rho} \Big|_{\mathbf{r} = \mathbf{R}_0}, \tag{9}$$

since in the saddle-point approximation, in the numerator of Equation (4), $\langle S(x, p) \rangle^{\mu}$ can be considered the "smooth"' function, and the determinant factors cancel out.

## 3. Some Exact Hydrodynamical Solutions and Polarization

In this section, we first specify and recapitulate the investigated hydrodynamical solutions, then give the analytical formulas for the polarization four-vector calculated from them. The equations of perfect fluid relativistic hydrodynamics utilized here are

$$
\begin{aligned}
(\varepsilon + p) u^{\nu} \partial_{\nu} u^{\mu} &= (g^{\mu\nu} - u^{\mu} u^{\nu}) \partial_{\nu} p & \text{(Euler equation)}, \\
(\varepsilon + p) \partial_{\mu} u^{\mu} &= -u^{\mu} \partial_{\mu} \varepsilon & \text{(energy conservation equation)}, \\
n \partial_{\mu} u^{\mu} &= -u^{\mu} \partial_{\mu} n & \text{(particle number/charge conservation)},
\end{aligned}
$$

and we specify the simple EoS:

$$\varepsilon = \kappa p, \tag{10}$$

where the notations are: $\varepsilon$, $p$, and $n$ are the energy density, pressure, and particle number density, respectively. Concerning the $n$ density: if it was assumed to be non-vanishing, we set the EoS as $p = nT$. However, the solutions presented below are valid also if $n = 0$ (i.e., if $\mu = 0$). Thus, the expressions for $n$ that we wrote up for the solutions can be regarded as supplemental to the solutions that work for $\mu = 0$.

We note that the simple analytic solutions of perfect fluid hydrodynamics that we utilize in this manuscript all assume this simple form of EoS, $\varepsilon = \kappa p$. Finding exact analytic relativistic solutions for a more complex equation of state is a daunting task (however, some simple developments have gradually been made in this direction, see e.g., Ref. [13]), but would be nevertheless required if one wants to use the methodology presented here to give constraints on the equation of state from the measured polarization effect of baryons. Such more general studies are beyond the scope of the present work, wherein we lay the groundwork for the analytic calculation of polarization. So we stick to the simple solutions (and their simple equation of state, $\varepsilon = \kappa p$) as discussed below.

We also note that there is recent development on taking the effect that polarization of the constituents of the fluid has on the fluid dynamics itself [14], along with some numerical calculations of how this modified hydrodynamical picture affects final state polarization [15]. We do not investigate this possibility here; we restrict ourselves to the simple and well-known basic equations written up above.

### 3.1. Hubble Flow

We do not go into the details about the method for finding or verifying that the solutions presented below are indeed solutions of the perfect fluid hydrodynamical equations; we refer back to the original publications of the solutions.

We investigate the Hubble-like relativistic hydrodynamical solution, first fully described in Ref. [8]. This solution has the following velocity, particle density, and temperature fields:

$$u^\mu = \frac{x^\mu}{\tau}, \qquad n = n_0 \left(\frac{\tau_0}{\tau}\right)^d, \qquad T = T_0 \left(\frac{\tau_0}{\tau}\right)^{d/\kappa}, \qquad (11)$$

where $\tau = \sqrt{t^2 - \mathbf{r}^2}$, and $\kappa$ is the inverse square speed of sound (constant in the case of this exact solution). The $\kappa = 3$ case corresponds to ultra-relativistic ideal gas, and $\kappa = 3/2$ corresponds to a non-relativistic gas; however, this solution is valid for any arbitrary constant $\kappa$ value[2]

To calculate the polarization four-vector, as of now we investigate the simplest case, the spherical symmetric expansion. For the freeze-out hypersurface, the $\tau = \tau_0 = \text{const.}$ hypersurface was chosen (which, in the case of the investigated solution, equals the constant temperature freeze-out hypersurface), and a given point of this hypersurface can simply be parametrized by the $\mathbf{r}$ coordinate three-vector, and the time coordinate on the hypersurface is $t(\mathbf{r}) \equiv \sqrt{\tau_0^2 + \mathbf{r}^2}$. The integration measure and the resulting expression for the Cooper–Frye formula can then be written as

$$\mathrm{d}^3 \Sigma_\mu = \frac{1}{t(\mathbf{r})} \begin{pmatrix} t(\mathbf{r}) \\ \mathbf{r} \end{pmatrix} \mathrm{d}^3 \mathbf{r} \qquad \Rightarrow \qquad E \frac{\mathrm{d}N}{\mathrm{d}^3 \mathbf{p}} = C_0 \int \mathrm{d}^3 \mathbf{r} \frac{Et(\mathbf{r}) - \mathbf{pr}}{t(\mathbf{r})} \exp\left(-\frac{Et(\mathbf{r}) - \mathbf{pr}}{T_0}\right). \qquad (12)$$

As we are discussing massive particles, this integral always exists. The $T_0$ constant (an arbitrary parameter of the solution) can simply be taken as the temperature at freeze-out, and we did so.

The position of the saddle-point ($\mathbf{R}_0$), as well as the second derivative matrix $M_{kl}$ is calculated as:

$$\partial_k \frac{Et - \mathbf{pr}}{T_0}\bigg|_{\mathbf{r} = \mathbf{R}_0} \overset{!}{=} 0 \quad \Rightarrow \quad \mathbf{R}_0 = \frac{\tau_0}{m} \mathbf{p}. \qquad M_{kl} \equiv -\partial_k \partial_l \frac{Et - \mathbf{pr}}{T_0}\bigg|_{\mathbf{r} = \mathbf{R}_0} = \frac{m}{T_0 \tau_0}\left(\delta_{kl} - \frac{p_k p_l}{E^2}\right). \qquad (13)$$

With this, we can get an approximation for the invariant single-particle momentum distribution:

$$\det \mathbf{M} = \frac{m^2}{E^2}\left(\frac{m}{T_0 \tau_0}\right)^3 \qquad \Rightarrow \qquad E \frac{\mathrm{d}N}{\mathrm{d}^3 \mathbf{p}} = \frac{n_0}{4} \sqrt{\frac{\pi \tau_0^3}{m T_0^3}} \exp\left(-\frac{\tau_0 m}{T_0}\right). \qquad (14)$$

The formula is independent of momentum. This was expected because this hydrodynamical solution (in the $\mathcal{V}(S) = 1$ case) is boost invariant.

To use (9) to determine the polarization four-vector in the hydrodynamical solution of the Hubble-flow, first we give the expression for the $\partial_\nu \beta_\rho$ derivative:

$$\partial_\nu \beta_\rho = \partial_\nu \left(\frac{r_\rho}{\sqrt{\tau_0^2 + r^2}\, T_0}\right) = \frac{g_{\nu\rho}}{\sqrt{\tau_0^2 + r^2}\, T_0} + \frac{r_\nu r_\rho}{(\tau_0^2 + r^2)^{3/2} T_0}. \qquad (15)$$

---

[2]　We note that a more general class of solutions is possible [8,9,16] in which the temperature and density fields are supplemented with an arbitrary $\mathcal{V}$ function of a "scaling variable" $S$:

$$n = n_0 \left(\frac{\tau_0}{\tau}\right)^d \mathcal{V}(S), \qquad T = T_0 \left(\frac{\tau_0}{\tau}\right)^{d/\kappa} \frac{1}{\mathcal{V}(S)},$$

and the $S$ variable is any function of $S_x$, $S_y$, and $S_z$:

$$S \equiv S(S_x, S_y, S_z), \quad \text{where} \quad S_x \equiv \frac{r_x^2}{\dot{X}_0^2 t^2}, \quad S_y \equiv \frac{r_y^2}{\dot{X}_0^2 t^2}, \quad S_z \equiv \frac{r_z^2}{\dot{X}_0^2 t^2}, \quad \text{for example:} \quad S = \frac{r_x^2}{\dot{X}_0^2 t^2} + \frac{r_y^2}{\dot{Y}_0^2 t^2} + \frac{r_z^2}{\dot{Z}_0^2 t^2}.$$

Here $\dot{X}_0$, $\dot{Y}_0$ and $\dot{Z}_0$ are arbitrary constants. In the given example, the $S = \text{const}$ surfaces are ellipsoids, and $\dot{X}_0$, $\dot{Y}_0$, $\dot{Z}_0$ are time derivatives of the principal axes of them.

Then, for the time component, we get:

$$\langle S(p)\rangle^0 = \frac{1}{8mT_0}\varepsilon^{0ikl}p_l\partial_i\beta_k\Big|_{\mathbf{r}=\mathbf{R}_0} = \frac{1}{8mT_0}\varepsilon_{ikl}p_l\left(\frac{g_{ik}}{\sqrt{\tau_0^2+r^2}T_0} + \frac{r_ir_k}{(\tau_0^2+r^2)^{3/2}T_0}\right)\Big|_{\mathbf{r}=\mathbf{R}_0} = 0, \quad (16)$$

as $\varepsilon^{0ikl}$ is antisymmetric, whereas $g_{ik}$ and $r_ir_k$ are symmetric to the change in the $i \leftrightarrow k$ indices.

Similarly for the spatial coordinates:

$$\langle S(p)\rangle^i = \frac{1}{8mT_0}\left(-\varepsilon_{ikl}p_l\partial_0\beta_k + \varepsilon_{ikl}p_l\partial_k\beta_0 - \varepsilon_{ikl}p_0\partial_k\beta_l\right)\Big|_{\mathbf{r}=\mathbf{R}_0} = 0. \quad (17)$$

In conclusion, the polarization four-vector in the spherical symmetric Hubble-flow is

$$\langle S(p)\rangle^\mu = \begin{pmatrix} 0 \\ \mathbf{0} \end{pmatrix}, \quad (18)$$

which is consistent with our expectations.

### 3.2. Rotating and Accelerating Expanding Solution

Another hydrodynamical solution of particular interest to us is a rotating and accelerating expanding solution, first written up in Ref. [10]. This solution has the following velocity, temperature, and particle density profiles:

$$\mathbf{v} = \frac{2t\mathbf{r}+\tau_0^2\mathbf{\Omega}\times\mathbf{r}}{t^2+r^2+\rho_0^2}, \qquad T = \frac{T_0\tau_0^2}{\sqrt{(t^2-r^2+\rho_0^2)^2+4\rho_0^2r^2-\tau_0^4(\mathbf{\Omega}\times\mathbf{r})^2}}, \qquad n = n_0\left(\frac{T}{T_0}\right)^3, \quad (19)$$

where $\rho_0$ and $\tau_0$ are arbitrary parameters, and $\mathbf{\Omega}$ is an arbitrary angular velocity three-vector that indicates the axis and magnitude of rotation. The $\rho_0$ parameter tells about the initial spatial extent of the expanding matter; however, the $\tau_0$ parameter is just there for the sake of consistency of physical units; in this way, the unit of $\mathbf{\Omega}$ is $c/\text{fm}$, as it should be for an angular velocity-like quantity[3], and $T_0$ is a temperature constant. In the case of $\mathbf{\Omega}=0$, we recovered an acceleratingly expanding but non-rotating spherically symmetric solution.

We note that in general $\mathbf{\Omega}\neq 0$, hence this solution has non-vanishing acceleration and rotation, as well as spatially non-trivial (i.e., not spherically symmetric) temperature distribution (and temperature gradient). In the $\mathbf{\Omega}\to 0$-limiting case (as noted above), the accelerating nature persists. However, in this case, the temperature distribution becomes spherically symmetric, and *at the same time*, the vorticity of the flow vanishes. In the case of this simple solution, we thus cannot choose the free parameters in a way to separately turn on and off these features, and thus cannot analytically disentangle the effects that these features have on the final state polarization. (Some numerical calculations of polarization, e.g., the one found in Ref. [17] state that different components of the polarization vector are influenced differently by these features of a realistic hydrodynamical expansion).

---

[3]　　Here we changed the notation of Ref. [10]. The rather unfortunate **B** notation used there is now written as $\tau_0^2\mathbf{\Omega}$.

Turning to the calculation of polarization, it is convenient to write up the investigated solution with the following notation:

$$\frac{u^\mu}{T} \equiv \beta^\mu = a^\mu + F^{\mu\nu}x_\nu + (x^\nu b_\nu)x^\mu - \frac{x^\nu x_\nu}{2}b^\mu, \tag{20}$$

$$\text{with} \quad a^\mu = \frac{\rho_0^2}{2T_0\tau_0^2}\begin{pmatrix}1\\0\end{pmatrix}, \qquad b^\mu = \frac{1}{T_0\tau_0^2}\begin{pmatrix}1\\0\end{pmatrix}, \qquad F_{0k} = F_{k0} = F_{00} = 0, \qquad F_{kl} = \varepsilon_{klm}\frac{\Omega_m}{2T_0}. \tag{21}$$

To calculate final state observables, we choose the constant proper time ($\tau_0 = $ const) hypersurface here as well. The solution itself allows for a re-scaling of the arbitrary constants in the formulas; just as in the previous case, here too we can treat the $T_0$ quantity as the temperature at freeze-out (at the **r** = 0 center of the expanding matter). We use the notation introduced in Equation (12) for the Maxwell–Boltzmann distribution. To derive the saddle-point for the calculation of the polarization four-vector, we shall use the expression of the invariant momentum spectrum:

$$E\frac{dN}{d^3\mathbf{p}} = C_0 \int d^3\mathbf{r}\left(E - \frac{\mathbf{pr}}{\sqrt{\tau_0^2+r^2}}\right)\exp\left\{-\frac{E(2r^2+\tau_0^2+\rho_0^2)-2\sqrt{\tau_0^2+r^2}\mathbf{pr}-\tau_0^2\mathbf{r}(\mathbf{p}\times\mathbf{\Omega})}{T_0\tau_0^2}\right\}. \tag{22}$$

This integral always exists (in the case of massive particles). In order to utilize the saddle-point integration method, we determine the position of the saddle-point ($\mathbf{R}_0$) and the second derivative matrix at the saddle-point:

$$\text{for } \mathbf{R}_0: \qquad \nabla\left\{-\frac{1}{T_0\tau_0^2}\left(E(2r^2+\tau_0^2+\rho_0^2) - 2\sqrt{\tau_0^2+r^2}\mathbf{rp} - \tau_0^2\mathbf{r}(\mathbf{p}\times\mathbf{\Omega})\right)\right\}\Big|_{\mathbf{r}=\mathbf{R}_0} \stackrel{!}{=} 0, \tag{23}$$

$$M_{kl} = \partial_k\partial_l\left\{\frac{1}{T_0\tau_0^2}\left(E(2r^2+\tau_0^2+\rho_0^2) - 2\sqrt{\tau_0^2+r^2}\mathbf{rp} - \tau_0^2\mathbf{r}(\mathbf{p}\times\mathbf{\Omega})\right)\right\}\Big|_{\mathbf{r}=\mathbf{R}_0}. \tag{24}$$

We leave the detailed calculations to Appendix A; the results are the following. The $\mathbf{R}_0$ saddle-point (for a given **p** momentum) is in the plane spanned by the **p** and $\mathbf{p}\times\mathbf{\Omega}$ vectors. In the following, we use the $\hat{\mathbf{p}} \equiv \mathbf{p}/p$ notation for the unit vector pointing in the direction of **p**. For the saddle-point, we get

$$\mathbf{R}_0 = \frac{\tau_0}{2p}\sqrt{\frac{E-m}{2m}}\sqrt{\tau_0^2(\hat{\mathbf{p}}\times\mathbf{\Omega})^2(E-m)^2 + 4p^2}\cdot\hat{\mathbf{p}} + \tau_0^2\frac{E-m}{2p}\cdot\hat{\mathbf{p}}\times\mathbf{\Omega}. \tag{25}$$

Concerning the second derivative matrix, we need it only for the calculation of the invariant momentum distribution, where its determinant is invoked. It turns out that this quantity is

$$\det M_{kl} = \frac{32m^2}{T_0^3\tau_0^6}(E+m)p. \tag{26}$$

Using this result, we get the invariant single-particle momentum distribution[4] as

$$E\frac{dN}{d^3\mathbf{p}} \propto \sqrt{\frac{\pi^3 T_0^3\tau_0^3}{32p(m+E)}}\exp\left(-\frac{E_{\text{eff}}}{T_0}\right), \quad \text{with} \quad E_{\text{eff}} = m + \frac{\rho_0^2 E}{\tau_0^2} + \frac{\tau_0^2}{4}\left(\mathbf{\Omega}^2 - (\hat{\mathbf{p}}\mathbf{\Omega})^2\right)(E-m). \tag{27}$$

---

[4]　This has not yet been calculated for this hydrodynamical solution.

Equivalently, by defining a "local slope" $T_{\text{eff}}$, the result can be expressed as

$$E\frac{\mathrm{d}N}{\mathrm{d}^3\mathbf{p}} \propto \sqrt{\frac{\pi^3 T_0^3 \tau_0^3}{32p(m+E)}} \exp\left(-\frac{E}{T_{\text{eff}}}\right), \quad \text{with} \quad T_{\text{eff}} = \frac{T_0}{\frac{m}{E}+\frac{\rho_0^2}{\tau_0^2}+\frac{\tau_0^2}{4}\left(\Omega^2-(\hat{\mathbf{p}}\mathbf{\Omega})^2\right)\left(1-\frac{m}{E}\right)}. \tag{28}$$

Proceeding to the polarization of the produced baryons, we calculated the derivative of the inverse temperature field for this solution from the form given in Equation (20), and then substituted it into the expression of the polarization, Equation (9). The result is

$$\partial_\nu\beta_\rho = F_{\rho\nu}+x^\alpha b_\alpha g_{\nu\rho}+x_\rho b_\nu-x_\nu b_\rho \quad \Rightarrow \quad \langle S(p)\rangle^\mu = \frac{1}{8m}\varepsilon^{\mu\nu\rho\sigma}p_\sigma\left(F_{\rho\nu}+x_\rho b_\nu-x_\nu b_\rho\right)\Big|_{\mathbf{r}=\mathbf{R}_0}. \tag{29}$$

(The second term was cancelled owing to the symmetry of $g_{\nu\rho}$ and the antisymmetry of $\varepsilon^{\mu\nu\rho\sigma}$, and $x^\mu$ is understood as the four-coordinate of the freeze-out hypersurface whose three-coordinate is the $\mathbf{r}=\mathbf{R}_0$ three-vector). Remembering the expression of the introduced $F^{\mu\nu}$ tensor and $b^\mu$ vector from Equation (20), in particular that of $F^{0k}=0$, and $b^k=0$, we got the following expressions for the the time-like and space-like components:

$$\langle S(p)\rangle^0 = -\frac{1}{8m}\varepsilon^{0klm}p_m(F_{kl}+x_lb_k-x_kb_l)\Big|_{\mathbf{r}=\mathbf{R}_0} = -\frac{1}{16m}\varepsilon_{klm}\varepsilon_{klq}p_m\frac{\Omega_q}{T_0} = \frac{1}{8m}\frac{\mathbf{p}\mathbf{\Omega}}{T_0},$$

$$\langle S(p)\rangle^k = \frac{1}{8m}\left(\varepsilon^{k0lr}p_r(F_{l0}+x_lb_0-x_0b_l)+\varepsilon^{kl0r}p_r(F_{0l}+x_0b_l-x_lb_0)+\varepsilon^{klr0}p_0(F_{rl}+x_rb_l-x_lb_r)\right)\Big|_{\mathbf{r}=\mathbf{R}_0}$$

$$= -\frac{1}{8m}\left(2b_0\varepsilon_{klm}x_lp_m+E\varepsilon_{klm}\varepsilon_{mlq}\frac{\Omega_q}{2T_0}\right)\Big|_{\mathbf{r}=\mathbf{R}_0} = \frac{1}{8mT_0}\left(E\mathbf{\Omega}-\frac{2}{\tau_0^2}\mathbf{R}_0\times\mathbf{p}\right)_k$$

$$= \frac{m\Omega_k+(E-m)\hat{p}_l\Omega_l\hat{p}_k}{8mT_0}.$$

Summarizing this result, the polarization four-vector for the investigated rotating and accelerating expanding solution is the following:

$$\langle S(p)\rangle^\mu = \frac{1}{8mT_0}\left(m\mathbf{\Omega}+\frac{\frac{\mathbf{p}\mathbf{\Omega}}{E-m}}{p^2}(\mathbf{\Omega}\mathbf{p})\mathbf{p}\right). \tag{30}$$

In the case of $\mathbf{\Omega}=0$, there is no rotation, and we got $\langle S(p)\rangle^\mu=0$. Thus, in this model, polarization is very transparently connected to the presence of rotation.

It is useful to transform the polarization four-vector into the rest frame of the particle. The result is[5], with (r.f. standing for "rest frame"):

$$\langle S(p)\rangle^\mu_{\text{r.f.}} = \begin{pmatrix} 0 \\ \mathbf{S}_{\text{r.f.}} \end{pmatrix}, \quad \text{where} \quad \mathbf{S}_{\text{r.f.}} = \frac{1}{8T_0}\mathbf{\Omega}. \tag{31}$$

We can also compute the helicity of the produced spin 1/2 particles in this solution from this formula (the $\mathbf{S}$ polarization vector is taken in the laboratory frame):

$$H := \hat{\mathbf{p}}\mathbf{S} = \frac{E}{8mT_0}\hat{\mathbf{p}}\mathbf{\Omega}. \tag{32}$$

---

[5]　The Lorentz matrix performing this boost transformation is the following (in usual 1+3 dimensional block matrix notation):

$$\Lambda^\mu{}_\nu = \begin{pmatrix} \cosh\chi & -\hat{p}_l\sinh\chi \\ -\hat{p}_k\sinh\chi & \delta_{kl}+(\cosh\chi-1)\hat{p}_k\hat{p}_l \end{pmatrix} = \frac{1}{m}\begin{pmatrix} E & -p_l \\ -p_k & m\delta_{kl}+\frac{E-m}{p^2}p_kp_l \end{pmatrix},$$

where $E$ and $p$ could be parametrized with the velocity parameter $\chi$ as $E=m\cosh\chi$ and $p=m\sinh\chi$, respectively. It indeed can be checked that this matrix takes the $(E,\mathbf{p})$ four-momentum vector into $(m,\mathbf{0})$, as it should.

## 4. Illustration and Discussion

In this section, we would like to illustrate our simple analytical results for the polarization vector. We use the same type of plots that were used to visualize some existing numerical simulations (e.g., those presented in Ref. [7]). We plot the components of the polarization vector with respect to the momentum components in the transverse plane (that is, w.r.t. momentum components $p_x$ and $p_y$). On Figure 1, we plot the polarization vector in the laboratory frame. For the sake of plotting, the mass of the $\Lambda$ baryon ($m_\Lambda c^2 = 1115$ MeV) was chosen. For the sake of this illustration, we chose a moderate value for the magnitude of the $\mathbf{\Omega}$ vector as $|\mathbf{\Omega}| = 0.1\,c/\text{fm}$.

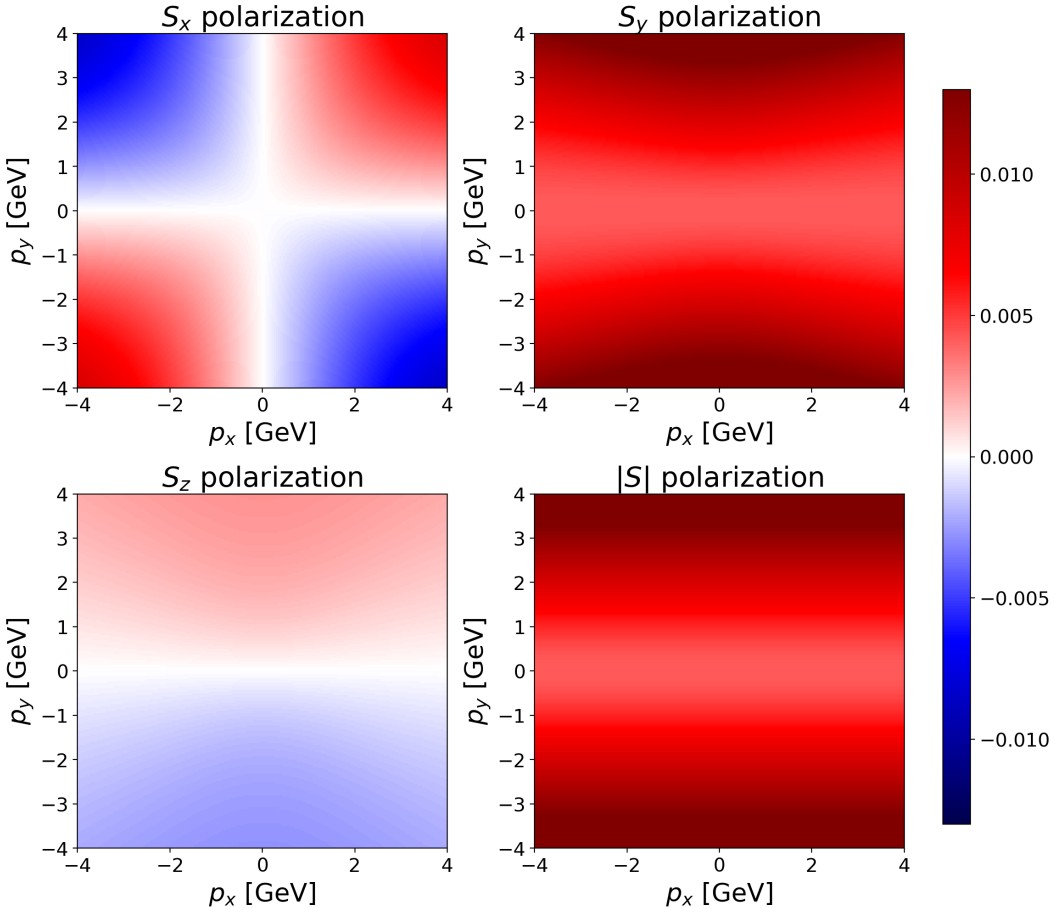

**Figure 1.** The components of the polarization four-vector in the rotating and accelerating expanding solution with respect to the momentum. Plots were made with the mass of the $\Lambda$ baryon ($m_\Lambda = 1115\,\text{MeV}/c^2$), and with $|\mathbf{\Omega}| = 0.1\,c/\text{fm}$.

In our case, as a special coincidence owing purely to the specific algebraic form of the presented analytic solution, it turned out that the polarization in the rest frame of the produced baryons was independent of momentum **p**; see Equation (31). This coincidence is expected to be relieved in the case of more involved (complicated) solutions (that are left for future investigations). Figure 2, nevertheless shows the value of the $S_y$ component in the baryon rest frame.

The helicity of the produced baryons (being proportional to the **pS** scalar product), however, *does* depend on the momentum, even in the case of our very simple solution. We plot it on Figure 3, with the same parameter values as in the foregoing two plots.

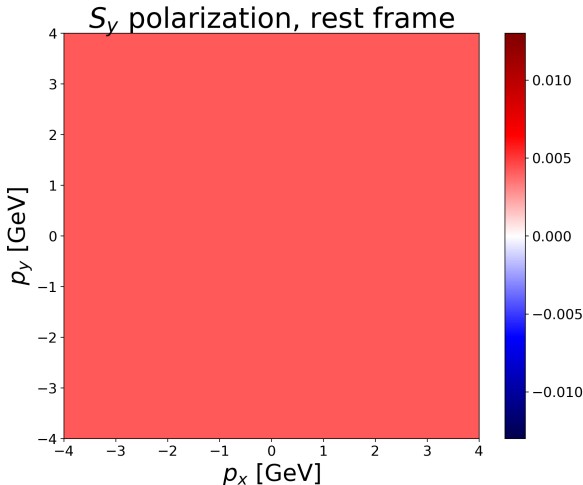

**Figure 2.** The only non-vanishing component of the polarization vector in the rest frame of the baryon is $S_y$ in the investigated simple solution; in this case, its value is uniquely determined by the magnitude of the $\Omega$ vector. More involved types of analytic solutions would yield some dependence on the momentum components, $p_x$ and $p_y$. For the plotted value of $S_y$ (a constant, as seen in the plot), the same input parameters were used as above: $m_\Lambda = 1115\,\text{MeV}/c^2$, and $|\Omega| = 0.1\,c/\text{fm}$.

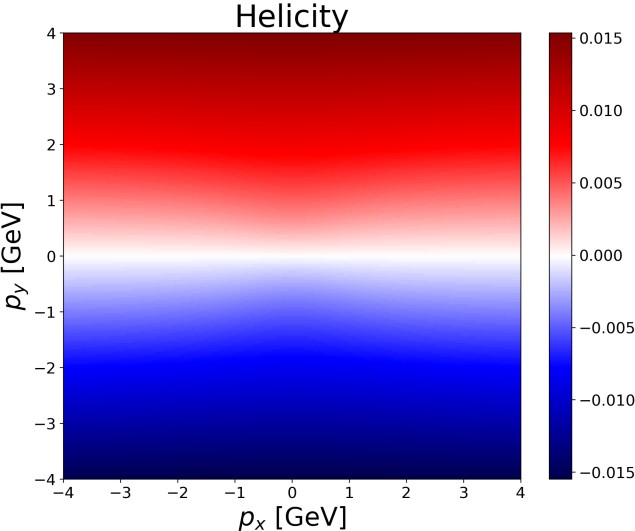

**Figure 3.** Helicity of the produced baryons calculated in the rotating and accelerating expanding solution. Parameter values as above: $m_\Lambda = 1115\,\text{MeV}/c^2$, $|\Omega| = 0.1\,c/\text{fm}$.

## 5. Summary and Outlook

In this paper, we gave the first analytical formulas for the polarization of baryons produced from a thermal ensemble corresponding to rotating and expanding exact hydrodynamical solutions. These arise as descriptions of the final state of non-central high energy heavy-ion collisions. We investigated two exact relativistic hydrodynamical solutions. One was the spherically symmetric Hubble flow (an overly simplistic one, the study of which can be regarded as a check of the methodology), in which the polarization turns out to be exactly zero (as is naturally expected from symmetry considerations). The other solution we investigated was one describing rotating and accelerating expansion. In this case, we obtained the first-ever analytical formulas that connected dynamical quantities of the expansion (i.e., magnitude of rotation, acceleration) with the observable final state polarization of spin 1/2 particles (baryons), which turned out to be non-zero in this case.

Our results are simple and straightforward. The calculations presented here yielded the first results in terms of exact formulas for the polarization. Nevertheless, many more solutions

(more involved ones), as well as more complicated final state parametrizations can be investigated in the future. The motivation is that the simple solution that we used here that yields non-zero polarization is one that features acceleration, temperature gradient, as well as vorticity. However, these cannot be tuned (or turned on and off) separately by a continuous change of the parameters of the solution. Solutions that allow this to be done are to be investigated in a later stage of this research effort. Such future studies are needed to disentangle the effects that rotation, acceleration, and temperature gradient have on the observable final state polarization of baryons produced in heavy-ion collisions. Such studies have the potential of a better understanding of what phenomenological implications polarization measurements (such as was recently done by the STAR experiment [1]) can have on the properties (such as the Equation of State) of the strongly coupled Quark Gluon Plasma produced in heavy-ion collisions.

**Author Contributions:** Conceptualization, M.C.; Investigation, B.B. and M.I.N.; Supervision, M.I.N. and M.C.; Visualization, B.B.; Writing-Original Draft, B.B. and M.I.N.; Writing-Review & Editing, M.I.N. and M.C.

**Funding:** Our research has been partially supported by the Hungarian NKIFH grants No. FK-123842 and FK-123959, the Hungarian EFOP 3.6.1-16-2016-00001 project. M. Csanád and M. Nagy was supported by the János Bolyai Research Scholarship of the Hungarian Academy of Sciences and the ÚNKP New National Excellence Program of the Hungarian Ministry of Human Capacities.

**Conflicts of Interest:** The authors declare no conflict of interest.

## Appendix A. Additional Calculations

Here we discuss some additional calculations used in Section 3.2 pertaining to the case of rotating and accelerating solution.

For a given momentum $\mathbf{p}$, the position of the saddle-point $\mathbf{R}_0$ (to be applied in the approximate calculation of the momentum spectrum and the polarization) was written up in Equation (25); we provide some additional details of the derivation of that formula here. The defining equation was Equation (23), of which the following equation for $\mathbf{R}_0$ is obtained:

$$4E\mathbf{R}_0 - 2\sqrt{\tau_0^2 + R_0^2}\,\mathbf{p} - \frac{2(\mathbf{p}\mathbf{R}_0)}{\sqrt{\tau_0^2 + R_0^2}}\mathbf{R}_0 - \tau_0^2(\mathbf{p}\times\mathbf{\Omega}) = 0, \tag{A1}$$

where $R_0^2 \equiv \mathbf{R}_0\mathbf{R}_0$. From this equation one readily sees that $\mathbf{R}_0$ must be a linear combination of $\mathbf{p}$ and the $\mathbf{p}\times\mathbf{\Omega}$ vector. We substitute this assumption into the equation above. We note that $\mathbf{p}$ and $\mathbf{p}\times\mathbf{\Omega}$ are orthogonal to each other, which leads to some intermediate simplifications, as well as enables us to rearrange the obtained condition into the following form:

$$\mathbf{R}_0 := \alpha\mathbf{p} + \beta\tau_0^2\mathbf{p}\times\mathbf{\Omega} \quad \Rightarrow \quad 2\left\{\left(2E - \frac{\alpha p^2}{A}\right)\alpha - A\right\}\mathbf{p} = \tau_0^2\left\{1 - 2\beta\left(2E - \frac{\alpha p^2}{A}\right)\right\}(\mathbf{p}\times\mathbf{\Omega}).$$

where we temporarily introduced the $A \equiv \sqrt{\tau_0^2 + \alpha^2 p^2 + \beta^2\tau_0^4(p^2\Omega^2 - (\mathbf{p}\mathbf{\Omega})^2)}$ notation. Because of the orthogonality of $\mathbf{p}$ and $\mathbf{p}\times\mathbf{\Omega}$, both sides here have to vanish identically, from which we get

$$A = \alpha\left(2E - \frac{\alpha p^2}{A}\right), \qquad\qquad 4E - \frac{2\alpha p^2}{A} = \frac{1}{\beta}. \tag{A2}$$

One divides these equations to obtain a simple relation, the substituting back one gets a quadratic equation for $\beta$, the solution of which is

$$\frac{\alpha}{\beta} = 2A \quad \Rightarrow \quad 4E - 4\beta p^2 = \frac{1}{\beta} \quad \Rightarrow \quad \beta = \frac{E}{2p^2} \pm \sqrt{\frac{E^2}{4p^2} - \frac{p^2}{4p^2}} = \frac{E \pm m}{2p^2}, \tag{A3}$$

where we used the $E^2 = p^2 + m^2$ relation. To find $\alpha$ we substitute this back into the expression of $A$:

$$\alpha = 2\beta A \quad \Rightarrow \quad \alpha^2 = 4\beta^2 \left\{ \tau_0^2 + \alpha^2 p^2 + \beta^2 \tau_0^4 (p^2 \Omega^2 - (\mathbf{p}\Omega)^2) \right\} \quad \Rightarrow \quad \alpha = 2\beta\tau_0 \sqrt{\frac{1 + \beta^2 \tau_0^2 (p^2 \Omega^2 - (\mathbf{p}\Omega)^2)}{1 - 4p^2\beta^2}}.$$

Using the above expression of $\beta$ (with the yet undetermined sign) we get $1 - 4p^2\beta^2 = -\frac{2m}{p^2}(m \pm E)$, and see that the expression for $\alpha$ will be valid only in the case when $1 - 4\beta^2 p^2 > 0$, thus conclude that the bottom sign is the proper choice. We thus arrive at the following expressions:

$$\beta = \frac{E - m}{2p^2}, \quad \alpha = 2\beta\tau_0 \sqrt{\frac{1 + \beta^2 \tau_0^2 (p^2 \Omega^2 - (\mathbf{p}\Omega)^2)}{1 - 4p^2\beta^2}} = \frac{\tau_0}{2} \sqrt{\frac{E - m}{2m}} \sqrt{\tau_0^2 (\hat{\mathbf{p}} \times \mathbf{\Omega})^2 (E - m)^2 + 4p^2}. \quad \text{(A4)}$$

From these formulas the expression of $\mathbf{R}_0$ shown in Equation (25) readily follows. The other ingredient in the saddle-point integration necessary for getting the momentum spectrum is the determinant of the second derivative matrix of the source function. Here we outline the main steps of the derivation of Equation (26). From Equation (24) the second derivative matrix itself turns out to be

$$M_{kl} = \frac{1}{T_0 \tau_0^2} \left\{ \left( 4E - \frac{2(\mathbf{pr})}{A} \right) \delta_{kl} - \frac{2}{A}(p_k r_l + r_k p_l) + 2(\mathbf{pr})\frac{r_k r_l}{A^3} \right\} \Bigg|_{\mathbf{r} = \mathbf{R}_0}, \quad \text{(A5)}$$

where we use the notation $A$ as above. We should use the expression of $\mathbf{R}_0$ as calculated above.

The determinant of this $\mathbf{M}$ matrix is the product of its eigenvalues. In our case the particular spatial directions are: $\mathbf{p}$, $\mathbf{p} \times \mathbf{\Omega}$, and the vector orthogonal to both these, that is, $\mathbf{p} \times (\mathbf{p} \times \mathbf{\Omega})$. One recognizes that the vector $\mathbf{p} \times (\mathbf{p} \times \mathbf{\Omega})$ is an eigenvector of the $\mathbf{M}$ second derivative matrix:

$$\mathbf{M}(\mathbf{p} \times (\mathbf{p} \times \mathbf{\Omega})) = \cdots = \frac{1}{\beta} \mathbf{p} \times (\mathbf{p} \times \mathbf{\Omega}). \quad \text{(A6)}$$

The corresponding eigenvalue is thus $1/\beta$. Owing to the symmetric nature of $\mathbf{M}$, the other two eigenvectors must be in the orthogonal complementer subspace of this vector, so they are linear combinations of $\mathbf{p}$ and $\mathbf{p} \times \mathbf{\Omega}$. Let us thus look for these eigenvectors in the form $\mathbf{a} = \mu\mathbf{p} + \nu\mathbf{r}$, with yet to be determined $\mu$ and $\nu$ coefficients. Substituting this expression, we get

$$\mathbf{M}\,\mathbf{a} = \lambda\mathbf{a} \quad \Rightarrow \quad \left( 4E - \frac{2(\mathbf{pR}_0)}{A} \right)\mathbf{a} - \frac{2}{A}\left( \mathbf{R}_0(\mathbf{ap}) + \mathbf{p}(\mathbf{aR}_0) \right) + 2(\mathbf{pR}_0)\frac{\mathbf{R}_0(\mathbf{aR}_0)}{A^3} = \lambda\mathbf{a}, \quad \text{(A7)}$$

where $\lambda$ is the eigenvalue (the values of which we are looking for). By substituting the assumed form of $\mathbf{a}$ and inferring the components of this equation in the $\mathbf{p}$ and $\mathbf{p} \times \mathbf{\Omega}$ directions, we get the following equation for the $\mu$ and $\nu$ coefficients:

$$\frac{2}{A^3} \begin{pmatrix} 2EA^3 - 2A^2\mathbf{pR}_0 & -A^2 R_0^2 \\ -A^2 p^2 + (\mathbf{pR}_0)^2 & 2EA^3 - 2A^2\mathbf{pR}_0 + R_0^2\mathbf{pR}_0 \end{pmatrix} \begin{pmatrix} \mu \\ \nu \end{pmatrix} = \lambda \begin{pmatrix} \mu \\ \nu \end{pmatrix}. \quad \text{(A8)}$$

We immediately infer the product of the two $\lambda_{1,2}$ eigenvalues as the determinant of this $2 \times 2$ matrix. Taking the third eigenvalue (calculated above) into account, after some simplifications, we indeed get the following expression for the determinant of the $\mathbf{M}$ matrix (the expression we used in Equation (26)):

$$\det \mathbf{M} = \left( \frac{1}{T_0 \tau_0^2} \right)^3 32 p m^2 (m + \sqrt{p^2 + m^2}). \quad \text{(A9)}$$

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
