# Peer review of "Polarized Baryon Production in Heavy Ion Collisions: An Analytic Hydrodynamical Study"

_universe, doi:10.3390/universe5050101_

Round 1

Reviewer 1 Report

I find the submitted paper suitable for publication in its present form.

Author Response

I find the submitted paper suitable for publication in its present form.

Response: We thank the Referee for this kind recommendation. We may note that the other referees requested slight modifications of the manuscript. We added two paragraphs: one around the beginning of Section 3. that elaborates oon the present situation of the possible EoS, stating that indeed there are possible future developments of exact solutions for more general EoS. We added another at the introduction of the rotating accelerating solution, after the beginning of Section 3.2 that explains that this solution does not allow for a separate tuning of acceleration, vorticity and T-gradient. We also slightly modified the Summary section to mirror these small changes. We hope that these slight modifications improve the quality of the manuscript and do not make the Referee reconsider the positive opinion (that we are grateful for).

Reviewer 2 Report

The authors present analytic results of the polarization of baryons produced in heavy ion collision based on the analytic solutions of perfect fluid hydrodynamics.
I think manuscript is clearly written and present useful results which
is the current interest regarding the polarization of lambda measured by the RHIC experiments.
I recommend publication of the present manuscript in the Universe.

For the improvements, it would be good to mention how you could study the different equation
of state (EoS) in this approach, since one of the motivation of this work is to study the sensitivities of EoS on the polarization. Current study only uses the simplest EoS.

Author Response

The authors present analytic results of the polarization of baryons produced in heavy ion collision based on the analytic solutions of perfect fluid hydrodynamics.
I think manuscript is clearly written and present useful results which
is the current interest regarding the polarization of lambda measured by the RHIC experiments.
I recommend publication of the present manuscript in the Universe.

Response: We would like to thank the referee for reviewing our manuscript and especially for the positive evaluation.

For the improvements, it would be good to mention how you could study the different equation
of state (EoS) in this approach, since one of the motivation of this work is to study the sensitivities of EoS on the polarization. Current study only uses the simplest EoS.

Response: We are grateful to the Referee for the kind recommendation. We modified the manuscript a little bit: we included a short paragraph about the possible future developments on the Equation of State right after the introduction of the hydro equations, at the beginning of Section 3. The situation is that indeed right now we have no knowledge of any exact analytic solution that admits a non-zero (and calculable!) polarization vector and yet is general enough to incorporate various types of general Equations of States. We state this in this paragraph (and mention one available model, which is however probably too simple to investigate vorticity in it), and hope that this slight modification improves our manuscript in the eyes of the Referee. Again, we thank for the kind consideration!

Reviewer 3 Report

The paper presents a first calculation of polarization of spin 1/2 particles (e.g. Lambda hyperons) in analytic solutions of relativistic hydrodynamics. The study is indeed interesting and relevant as the final expression for the polarization relates it analytically to the parameters of the hydrodynamic expansion, therefore it can be also useful as a benchmark for more complex hydrodynamic scenarios. It absolutely deserves to be published.
 I would like the authors to clarify one question. The analytical solution which provides nonzero polarization (Sect. 3.2) is both accelerating and rotating. It has been shown in a publication by other group [Nucl. Phys. A 982 (2019), 519-522, arXiv:1811.00322] that in a numerical 3D hydrodynamic expansion and using the same modified Cooper-Frye formula, the different components of polarization are induced by different mechanisms: whereas the component in the direction of the total angular momentum is induced by vorticity (therefore, by rotation itself), the component in the longitudinal direction is induced by acceleration and gradients of temperature (the latter in the case of ideal fluid is again proportional to the acceleration). Whereas with the analytical solution from Eq. (31) it seems that all the components of polarization are proportional to the angular velocity \Omega only. Therefore could you please clarify if the acceleration itself enters the expression for polarization - or perhaps the acceleration and rotation are related to each other in the given analytic hydro solution?

Author Response

The paper presents a first calculation of polarization of spin 1/2 particles (e.g. Lambda hyperons) in analytic solutions of relativistic hydrodynamics. The study is indeed interesting and relevant as the final expression for the polarization relates it analytically to the parameters of the hydrodynamic expansion, therefore it can be also useful as a benchmark for more complex hydrodynamic scenarios. It absolutely deserves to be published.

Response: We thank the Referee for the review and positive evaluation of our work.

I would like the authors to clarify one question. The analytical solution which provides nonzero polarization (Sect. 3.2) is both accelerating and rotating. It has been shown in a publication by other group [Nucl. Phys. A 982 (2019), 519-522, arXiv:1811.00322] that in a numerical 3D hydrodynamic expansion and using the same modified Cooper-Frye formula, the different components of polarization are induced by different mechanisms: whereas the component in the direction of the total angular momentum is induced by vorticity (therefore, by rotation itself), the component in the longitudinal direction is induced by acceleration and gradients of temperature (the latter in the case of ideal fluid is again proportional to the acceleration). Whereas with the analytical solution from Eq. (31) it seems that all the components of polarization are proportional to the angular velocity \Omega only. Therefore could you please clarify if the acceleration itself enters the expression for polarization - or perhaps the acceleration and rotation are related to each other in the given analytic hydro solution?

Response: We whole-heartedly accept the suggestion that was made about clarifying one point about the used relativistic hydro solution. We modified the manuscript to include this detail (added a paragraph between Eqs. 18 and 19, as well as slightly re-written the Summary section, as well as included the mentioned reference). Indeed the case is that our solution does not allow for a separate tuning of the acceleration (which is present even in the non-rotating limiting case of the solution) and that of the temperature gradient and the vorticity (these become spherically symmetric and zero, respectively, in the Omega=0 limiting case). So our formulas presented here do not yet allow for an analytic insight into the separate effects of these features. As for overcoming this drawback of the present calculation, we can only hope that more general calculations that assume more general solutions and/or final state parametrizations (which we already started as a follow-up study to the presented work) will be able to really disentangle the effects of rotation, temperature gradient and vorticity in an analytic manner, hopefully supplementing the interesting numerical calculations that are done towards this end.

Summarizing, we gratefully accepted the proposed modification, and hope that the revised manuscript will be met with acceptance on the part of the Referee. And we thank again the kind recommendation.